# An Assessment of Human Opportunistic Pathogenic Bacteria on Daily Necessities in Nanjing City during Plum Rain Season

**DOI:** 10.3390/microorganisms12020260

**Published:** 2024-01-26

**Authors:** Xiaowei Yu, Yifan Yin, Zuoyou Wu, Hui Cao

**Affiliations:** 1Key Laboratory of Agricultural Environmental Microbiology, Ministry of Agriculture and Rural Affairs, College of Life Sciences, Nanjing Agricultural University, Nanjing 210095, China; 2021116038@stu.njau.edu.cn (X.Y.); 2018216014@njau.edu.cn (Y.Y.); 2Sir Run Run Shaw Hospital, Nanjing Medical University, Nanjing 211112, China

**Keywords:** plum rain season, daily necessities, human opportunistic pathogenic bacteria, co-occurrence network, community assembly, health risks

## Abstract

The plum rain season is a special climatic phenomenon in east Asia, which is characterized by persistent rainfall, a high temperature, and humidity, providing suitable environmental conditions for certain pathogenic bacteria, thus increasing the incidence of respiratory, gastrointestinal, and urinary diseases. However, studies on human opportunistic pathogenic bacteria communities during the plum rain season are still limited. In this study, the characteristics of human opportunistic pathogenic bacterial communities on daily necessities during the non-plum and plum rain seasons were investigated using high-throughput sequencing technology. The results revealed that the relative abundance of human opportunistic pathogenic bacteria was higher in the plum rain season (cotton cloth: 2.469%, electric bicycles: 0.724%, rice: 3.737%, and washbasins: 5.005%) than in the non-plum rain season (cotton cloth: 1.425%, electric bicycles: 0.601%, rice: 2.426%, and washbasins: 4.801%). Both temperature and relative humidity affected human opportunistic pathogenic bacterial communities. Stochastic processes dominated the assembly process of human opportunistic pathogenic bacterial communities, and undominated processes prevailed. The stability of the co-occurrence network was higher in the non-plum rain season than that in the plum rain season. In addition, the proportion of deterministic processes showed the same trend as the complexity of the co-occurrence network.

## 1. Introduction

Plum rain is one of the typical climatic phenomena in the subtropical monsoon region, which refers to the persistent rainfall that occurs in June and July every year in the Yangtze-Huaihe River Basin, southern Taiwan, the Liaodong Peninsula of China, and southern Japan [1,2]. The plum rain season is characterized by high temperatures, humidity, and inherent repetitiveness, during which, precipitation accounts for 20–50% of the total annual precipitation [3], which provides suitable environmental conditions for certain bacteria. Direct exposure to pathogens can lead to infections, while exposure to microbial communities in different environments may modulate the human immune system [4]. Seasonal patterns of many infectious diseases and immune-mediated disorders of public health importance have been identified. The incidence of some common diseases follows a seasonal pattern. Therefore, understanding how the plum rain season influences microbial communities can provide valuable insights into the epidemiology of seasonal infectious diseases [5].

The climate during the plum rain season is favorable for the growth of certain bacteria, thus leading to a high incidence of various diseases. For diseases with a high incidence in the plum rain season, pathogenic bacteria are usually isolated and purified from lesions and identified after microscopic observation of cultures and serological testing [6]. The incidence of melioidosis (more than 70% presenting as pulmonary melioidosis) has been reported to be associated with the presence of heavy rainfall [7], and humans become infected with melioidosis through direct contact with, ingestion of, or inhalation of the Gram-negative bacterium *Burkholderia pseudomallei* in contaminated soil or dust [8,9,10]. Some studies have found an increased incidence of allergic rhinitis in children due to the humidity and temperature conditions favorable for dust mite reproduction during the plum rain season [11]. In Anhui Province, heavy precipitation during the 2016 plum rain season caused severe flooding events, which led to a significant increase in the risk of infectious diarrhea, particularly in women and children [12]. Statistics on the number of patients hospitalized for urinary tract diseases have been compiled and it was found that urinary tract diseases also exhibit seasonal trends, with higher hospitalization rates during the plum rain season [13]. In addition, a common skin disease, cellulitis, usually caused by group A Streptococci, is more prevalent during the plum rain season [14]. The above studies found that certain diseases caused by bacteria are highly prevalent during the plum rain season, but did not investigate the microbial community characteristics involved.

Most of the current studies on the plum rain season have focused on meteorology [3,15] and hydrology [16,17], with relatively few studies involving microbial variations. A previous study focused on airborne bioaerosols in university student dormitories during the plum rain season and found that there was no significant difference in bacterial aerosol concentrations between the morning and afternoon, and that airborne particulate matter concentration and relative humidity were negatively correlated with bacterial aerosol concentrations, whereas temperature was positively correlated with bacterial aerosols [18]. In addition, a study investigated airborne bioaerosols in commuter trains throughout the year in Taiwan and found that CO_2_ concentration was negatively correlated with bacterial concentration during the summer rainy season, and that bacteria in different compartments fluctuated, affected by relative humidity [19]. In addition, it was also found that bacterial viability was highest in the rain of the plum rain fronts compared to other precipitation, and that bacteria involved in processes in the clouds may be an important source of bacteria in rainwater [20]. However, the above studies only focused on the number of culturable microorganisms and ignored the non-culturable microorganisms, without exploring the characteristics of the bacterial communities therein in depth; thus, this needs to be further expanded and explored in detail.

In this study, we collected microbial samples from cotton cloth, electric bicycles, rice, and washbasins before, during, and after the rainy season and used high-throughput sequencing to investigate the characteristics of human opportunistic pathogenic bacterial communities on daily necessities. We hypothesized that (1) the plum rain season affects the relative abundance of human opportunistic pathogenic bacteria and the relative abundance of human opportunistic pathogenic bacteria is higher in the plum rain season than in the non-plum rain season; (2) the plum rain season alters the diversity of human opportunistic pathogenic bacterial communities; (3) the structure of the co-occurrence network of these bacterial communities is changed during the plum rain season; and (4) both temperature and relative humidity affect human opportunistic pathogenic bacterial communities. These results may provide a theoretical reference for the health risks posed to the human living environment during the plum rain season and call for better public health management during the plum rain season.

## 2. Materials and Methods

### 2.1. Study Site

The study site, situated in Nanjing (118°22′ E–119°14′ E, 31°14′ N–32°37′ N) within Jiangsu Province in eastern China, features a humid north subtropical climate marked by distinct seasons and abundant precipitation. With an average of 117 days of annual rainfall averaging 1106.5 mm and a frost-free span lasting 237 days, this locale is characterized by the plum rain season—a continuous rainy period prevailing from early June to mid-July each year.

### 2.2. Experimental Design and Sampling

The experiment was conducted from 14 June to 18 July 2022, including two different stages, the non-plum and plum rain season. The specific sampling dates were 20 June, 27 June, 4 July, 11 July, and 18 July. Daily temperature, relative humidity, and precipitation data were also recorded from 14 June to 18 July, and the temperature and relative humidity of the seven days prior to the sampling time point were combined and processed as the average temperature and relative humidity at the sampling time point. Additionally, historical information about the frequency and intensity of plum rains over the past decade in Nanjing was available (http://js.cma.gov.cn/dsjwz/njs/, accessed on 18 July 2022), along with information on the plum rain season and the climate characteristics throughout all seasons of the year in 2022 (https://www.stats.gov.cn/, accessed on 6 January 2024).

Four daily necessities, respectively, cotton cloth (C), electric bicycles (E), rice (R), and washbasins in dormitories (W), selected from the four aspects of clothing, transportation, food, and housing that are closely related to human activities, were sampled. Cotton cloth and rice are located in an open environment, washbasins are located on dormitory balconies with no barriers and direct access to the outdoors, and electric carts are located in a completely outdoor environment, with all daily necessities being directly exposed to the atmosphere. For each instance, approximately 0.28 g of cotton cloth and 0.5 g of rice were sampled. Sterile cotton swabs were dampened with sterile water and used to meticulously swab within a 5 cm × 5 cm area for 5 to 10 s to ensure the comprehensive collection of surface microorganisms from the electric bicycles and washbasins. Three replicates of each sample were taken and a total of 60 samples were collected during the 29-day sampling period.

### 2.3. DNA Extraction, PCR Amplification, and Sequencing

Genomic DNA was extracted by using the MolPure^®^ Soil DNA Kit (Yeasen Biotechnology Co., Shanghai, China) in accordance with the manufacturer’s instructions. The concentration and purity were measured using the NanoDrop 2000 (Thermo Fisher Scientific, Waltham, MA, USA). Then, the extracted genomic DNA was stored at −80 °C for a PCR and sequencing analysis.

The 16S rRNA gene was amplified by the V3-V4 region with the universal primers 341F (5’-CCTACGGGNGGCWGCAG-3’) and 805R (5’-GACTACHVGGGTATCTAATCC-3’) [21]. The 16S rRNA gene V3-V4 PCR, containing 12.5 µL of Phusion^®^ Hot Start Flex 2 X Master Mixart Version (NEB, M0536L, Beijing, China), 2.5 µL of Forward Primer (1 μM), 2.5 µL of Reverse Primer (1 μM), 50 ng of template DNA, and 25 µL of ddH2O added to the total system for 60 samples, was amplified by thermocycling: 30 s at 98 °C for initialization; 35 cycles of 10 s denaturation at 98 °C, 30 s annealing at 54 °C and 45 s extension at 72 °C; followed by 10 min final elongation at 72 °C.

The target fragments were recovered using the AxyPrep PCR Cleanup Kit (Corning Life Sciences, Corning, NY, USA). The PCR product was further purified using the Quant-iT PicoGreen dsDNA Assay Kit (Thermo Fisher Scientific, MA, USA). The library was quantified on the Promega QuantiFluor fluorescence quantification system. The pooled library was loaded on the Illumina platform (Illumina Inc., San Diego, CA, USA) using a paired-end sequencing protocol (2 × 250 bp).

### 2.4. Processing of Sequence Analysis

Paired-end reads were assigned to specific samples based on their distinctive barcodes. Subsequently, the barcode and primer sequences were trimmed from the paired-end reads, which were then merged using FLASH (v1.2.8). The quality filtering of raw reads was conducted under predefined conditions to obtain high-quality, clean tags in accordance with fqtrim (v0.94). Chimeric sequences were eliminated using the Vsearch software (v2.3.4). Sequences with a 97% similarity were clustered into Operational Taxonomic Units (OTUs), employing the Usearch software (https://www.drive5.com/usearch/, accessed on 11 November 2022). All OTUs were assigned a taxonomic category with a confidence threshold of 0.8 using the Ribosomal Database Program (RDP) classifier. The Silva database was used by the RDP classifier, which predicts taxonomic classifications from kingdom to species level.

### 2.5. Data Analysis and Statistics

Alpha diversity (α diversity), including the Shannon index and Chao1 index, was calculated using the “diversity” function in the “Vegan” package of the software RStudio (version 4.2.0). As to whether there were statistically significant differences in the diversity of the samples at different stages, one-way analysis of variance (ANOVA) and Duncan’s test were applied. Statistical analyses were conducted using software SPSS version 25.0. Principal coordinate analysis (PCoA) was employed to evaluate the distribution patterns of the microorganisms during different plum rain stages. The analysis was based on beta-diversity calculated using the Bray–Curtis distance, facilitated by the “ade4” package in Rstudio (version 4.2.0). To identify human opportunistic pathogenic bacteria, sequencing data were compared to the Enhanced Infectious Disease Database [22]. Subsequent pie charts depicting human opportunistic pathogens were generated using Origin 2023. The assembly processes of human opportunistic pathogenic bacterial communities were constructed using the “ses.comdistnt” function in the “MicEco” package [23], with the beta mean nearest taxon distance (βMNTD) metric used to determine turnover in the phylogenetic structure of the communities. Meanwhile, the stochastic and deterministic ecological processes of human opportunistic pathogenic bacterial communities were evaluated through a null model analysis [23]. The β-nearest taxon index (βNTI) was used to evaluate the deviation between the mean of the null βMNTD and observed βMNTD, expressed in units of standard deviations [24]. The assembly process of human opportunistic pathogenic bacterial communities was quantified using the “iCAMP” R package [25]. Meanwhile, the influences of the variable selection and homogeneous selection fractions were determined according to thresholds of βNTI values > 2 and <−2, respectively. The relative influence of dispersal limitation was quantified as a pairwise comparison between |βNTI| < 2 and RCbray > 0.95, whereas |βNTI| < 2 and RCbray < 0.95 was used to estimate the influence of undominated processes [26]. The construction of the environmental preferences of human opportunistic pathogenic bacterial communities was informed by established research methodologies [27]. Microbial networks were created to establish interaction networks for both non-plum rain and plum rain seasons. Each network consisted of 500 OTUs, encompassing both human opportunistic pathogenic and non-pathogenic bacteria. The Spearman’s rank correlation coefficients were computed using the R package “psych”. Correction for the analysis of the network was conducted for correlation coefficients (*p* < 0.05 and r > 0.6). The resulting network graphs were visualized, and topological properties were calculated using the Gephi software (version 0.9.2).

## 3. Results

### 3.1. Characteristics in Temperature and Relative Humidity during Plum Rain Season

The duration and precipitation of the 2022 Nanjing plum rain season were within the record range compared to the plum rain of the past decade (Table 1 and Appendix A). In addition, the average temperature, average relative humidity, and precipitation were higher in the plum rain season than in other seasons, except that the average temperature was slightly lower than the summer average temperature (Appendix A). Notably, during the 2022 plum rain season, the average minimum temperature (A.Tmin), average temperature difference (A.itd), and average relative humidity (A.RH) exhibited significant increases compared to the period before the plum rain season. Moreover, A.RH demonstrated a marked rise during the plum rain season in contrast to the non-plum rain season, while A.itd experienced a significant decrease. Conversely, the average maximum temperature (A.Tmax) registered a notable increase after the plum rain season (Table 1). In conclusion, the plum rain season in Nanjing for the year 2022 can be characterized as being typical and within the normal range, exhibiting the standard characteristics of the plum rain pattern.

### 3.2. Variations in Community Diversity and Composition of Human Opportunistic Pathogenic Bacteria on Daily Necessities

A total of 2,943,769 high-quality reads were obtained from 60 samples after filtering low-quality reads and chimaeras and trimming the adapters, primers, and barcodes.

The α diversity of human opportunistic pathogenic bacterial communities was calculated using the Shannon index and Chao1 index (Figure 1A,B). On all daily necessities, the Shannon index was lower during the plum rain season than that during the non-plum rain season, while the Chao1 index of human opportunistic pathogenic bacterial communities on electric bicycles and rice was lower during the plum rain season than that during the non-plum rain season, and the opposite results were observed for the Chao1 index of human opportunistic pathogenic bacterial communities on cotton cloth and washbasins. A principal coordinate analysis (PCoA) based on Bray–Curtis dissimilarity indicated significant differences in human opportunistic pathogenic bacterial community compositions (*p* = 0.001, Appendix A). The PCoA axis 1 explained 39.47% and axis 2 accounted for 14.7% of the total variance, effectively separating the human opportunistic pathogenic bacterial communities of cotton cloth, rice, and washbasins. In addition, the human opportunistic pathogenic bacterial communities after plum rain were clustered separately from the other stages (Appendix A).

A total of 11 human opportunistic pathogens were identified (Figure 1C), which can cause three types of diseases: Stenotrophomonas maltophilia, Pseudomonas aeruginosa, Klebsiella pneumoniae, Mycobacterium tuberculosis, Legionella pneumophila, and Haemophilus influenzae, which cause respiratory diseases, were found in the highest relative abundance on washbasins (non-plum rain season: 4.752% and plum rain season: 4.864%); Listeria monocytogenes, Bacillus cereus, and Staphylococcus aureus, which cause gastrointestinal diseases, had the highest relative abundance on rice (non-plum rain season: 0.923% and plum rain season: 1.298%); and Enterococcus faecium and Proteus mirabilis, which cause urinary tract diseases, had the highest relative abundance on cotton cloth (non-plum rain season: 0.168% and plum rain season: 1.375%). The relative abundance of human opportunistic pathogenic bacteria on cotton cloth, electric bicycles, rice, and washbasins in the non-plum rain season was 1.425%, 0.601%, 2.426%, and 4.801%, respectively, and the relative abundance of human opportunistic pathogenic bacteria on cotton cloth, electric bicycles, rice, and washbasins in the plum rain season was 2.469%, 0.724%, 3.737%, and 5.005%, respectively, on all daily necessities, all of which had a higher relative abundance of human opportunistic pathogenic bacteria during the plum season than the non-plum rain season (Figure 1C). In addition, the highest number of human opportunistic pathogenic bacteria species was found on cotton cloth and the highest relative abundance of human opportunistic pathogenic bacteria was found on the washbasins.

### 3.3. Influence of Environmental Factors on Human Opportunistic Pathogenic Bacteria on Daily Necessities

Human opportunistic pathogenic bacteria on washbasins showed the highest susceptibility to the combined effects of temperature and relative humidity, followed by human opportunistic pathogenic bacteria on cotton cloth, electric bicycles, and rice (Figure 2A–D). For human opportunistic pathogenic bacteria causing gastrointestinal diseases (Appendix A), *Staphylococcus aureus* on rice was most affected by temperature and relative humidity, followed by *Staphylococcus aureus* on cotton, and *Staphylococcus aureus* on washbasins, which was only affected by temperature. For human opportunistic pathogenic bacteria causing respiratory diseases (Appendix A), *Stenotrophomonas maltophilia* on cotton cloth, rice, and washbasins all had a high correlation with relative humidity, and *Klebsiella pneumoniae* on cotton cloth, electric bicycles, and washbasins were all affected by a combination of temperature and relative humidity, with only *Pseudomonas aeruginosa* on cotton cloth and washbasins having a correlation with temperature. For human opportunistic curing bacteria causing urinary tract diseases (Appendix A), *Enterococcus faecium* on cotton cloth, rice, and washbasins were positively affected by temperature, whereas *Enterococcus faecium* on electric bicycles was negatively affected by temperature and positively affected by relative humidity, probably because electric bicycles were completely in the outdoor environment.

### 3.4. Assembly Processes of Human Opportunity Pathogenic Bacteria on Daily Necessities

To differentiate between deterministic and stochastic processes in community assembly, the βNTI was calculated for the plum rain and non-plum rain seasons, and the contributions of deterministic and stochastic processes were obtained (Figure 3A). Regardless of the season, the assembly of human opportunistic pathogens on cotton cloth, electric bicycles, rice, and washbasins was predominantly driven by stochastic processes; additionally, the assembly of human opportunistic pathogenic bacteria on electric bicycles and washbasins exhibited a higher proportion of deterministic processes compared to that on cotton cloth and rice.

The assembly processes of human opportunistic pathogenic bacteria were comprehensively analyzed using quantitative methods (Figure 3B). The stochastic process was composed of undominated and homogeneous dispersal, and undominated accounted for the vast majority. Deterministic processes consisted only of variable selection. Undominated contributed the most to all community assembly processes, followed by variable selection and then homogeneous dispersal.

### 3.5. An Overview of Constructed Human Opportunistic Pathogenic Bacterial Co-Occurrence Networks

Distinct differences in the co-occurrence networks of human opportunistic pathogens on daily necessities during the plum rain and non-plum rain seasons were observed (Figure 4A). A comparison of nodes and edges data between the plum rain and non-plum rain seasons revealed that the bacterial networks on cotton cloth and electric bicycles were more complex within the plum rain season, and the bacterial community networks on rice and washbasins were more complex in the non-plum rain season (Table 2). In addition, associations between human opportunistic pathogens and other bacteria on cotton cloth, rice, and washbasins were more abundant in the non-plum rain season than in the plum rain season.

Network stability was assessed by observing natural connectivity when 50% of the nodes were removed (Figure 4B). The natural connectivity of the bacterial co-occurrence networks of cotton cloth, electric bicycles, rice, and washbasins with 50% of the nodes removed were 0.175 and 0.103, 0.203 and 0.165, 0.130 and 0.026, and 0.171 and 0.091 within the non-plum and plum rain seasons, respectively, implying that the bacterial co-occurrence networks during the non-plum rain season consistently showed a higher stability. Meanwhile, the results of the network modularity and negative edges confirmed this finding (Table 2).

## 4. Discussion

### 4.1. Relative Abundance of Human Opportunistic Pathogens Increased but Community Diversity Decreased during the Plum Rain Season

In this study, we found that cotton cloth had the highest number of species of human opportunistic pathogens, probably attributed to the fact that cotton cloth consists of cotton fibers, which have a high adsorption capacity for many species of bacteria from the surrounding environment. In addition, the relative abundance of human opportunistic pathogens on daily necessities was higher in the plum rain season than in the non-plum rain season, which may be attributed to the fact that certain human opportunistic pathogens prefer humid environments, such as Listeria monocytogenes [28], and previous studies have also shown that humid environments can increase the relative abundance of human opportunistic pathogens in the environment [29]. Compared to the environments in which the other daily necessities were located, washbasins were in more humid environments, which may explain the higher relative abundance of opportunistic human pathogenic bacteria on the washbasins than on the other three daily necessities.

The Shannon index is one of the indicators used to estimate community diversity [30], while the Chao1 index indicates species richness and is sensitive to changes in rare species [31]. In previous studies, community diversity decreased when strong competitors dominated, i.e., the Shannon index decreased [32]. In this study, the relative abundance of the dominant species *Stenotrophomonas maltophilia* and *Staphylococcus aureus* was elevated during the plum rain season compared to the non-plum rain season, resulting in a decrease in the community’s Shannon index. In addition, rare species usually account for a large proportion of species richness [33], and the Chao1 index of a community increases when rare species are present. The presence of *Haemophilus influenzae* on cotton cloth and *Staphylococcus aureus* on washbasins during the plum rain season may have beem due to their greater preference for moist environments [34], which would lead to an increase in the Chao1 index.

### 4.2. Both Temperature and Relative Humidity Affect Human Opportunistic Pathogenic Bacterial Communities

*Staphylococcus aureus* can form biofilms on inanimate surfaces, and, in a previous study, *Staphylococcus aureus* concentration was found to be correlated with temperature [35], whereas in the present study, *Staphylococcus aureus* on daily necessities, except for electric bicycles, was correlated with temperature and humidity, which may be due to the fact that the above study investigated the effect of environmental changes on *Staphylococcus aureus* in the air, whereas the present study investigated the effect of environmental changes on *Staphylococcus aureus* on daily necessities. In addition, one study found that the incidence of foodborne illnesses caused by *Bacillus cereus* increased during the plum rain season in June and July [36], which is consistent with the findings of the present study that *Bacillus cereus* on washbasins was positively affected by temperature. For human opportunistic pathogens that cause respiratory diseases, previous studies have found that the incidence of *Klebsiella pneumoniae* was significantly correlated with changes in temperature and humidity, with a 7% increase in the prevalence of the Gram-negative pathogen *Klebsiella pneumoniae* in hospitals for every 5 °C increase in temperature [37], which is consistent with the results of the present study, where temperature had a positive effect on *Klebsiella pneumoniae* on washbasins.

The positive effect of temperature on the pathogenic bacteria of urinary tract diseases on cotton cloth, rice, and washbasins was due to the fact that *Enterococcus faecium* is a human parasite that grows more rapidly at 37 °C [38], a finding suggesting that temperature has a positive effect on *Enterococcus faecium*, given that the temperature conditions in the present study were consistently below this optimal value. In addition, relative humidity is another important factor in the growth of *Enterococcus faecium*.

Since indoors and outdoors can be affected by different environmental conditions such as lighting, ventilation, I/O ratio, PM size, and many other factors [39], these may lead to different responses of indoor and outdoor bacterial communities located in completely outdoor environments to environmental changes, as, in this study, the *Enterococcus faecium* on the electric bicycles that were in a completely outdoor environment were negatively affected by temperature as well as positively affected by relative humidity.

### 4.3. Stochastic Processes Dominate Human Opportunistic Pathogenic Bacterial Communities on Daily Necessities, and Human Exposure as a Selection Pressure

Community assembly is the process of species colonization and interaction that establishes and maintains local communities through successive repeated migrations from a regional species pool [40]. The process of community assembly determines patterns of species distribution and abundance, influenced by both deterministic and stochastic processes [23]. In both the non-plum rain and plum rain seasons, human opportunistic pathogen communities were dominated by stochastic processes, and undominated processes (ecological drift) predominated among them. Ecological drift is characterized by fluctuations in microbial abundance and changes in population size (immigration, mortality, and random births) [41]. Therefore, we suggest that a stochastic balance between the deaths and births of human opportunistic pathogens (ecological drift) can generate new community structures and assemblies on daily commodities. This was consistent with previous findings that drift was most important when selection was weak, α-diversity was low, and total community membership was low [42]. Notably, previous studies have revealed that anthropogenic selection pressures are similar to natural ecological selection, promoting more deterministic processes in the form of selection [43,44]. In this study, the electric bicycles and the washbasins were exposed to a higher frequency of human activities than the cotton and rice, which, consequently, accounted for a higher percentage of deterministic processes in the assembly of their communities than that of cotton cloth and rice. However, the proportion of homogeneous dispersal on the electric bicycles in the non-plum rain season was higher than that in the plum rain season, and the homogeneous dispersal led to a more similar composition [45], which was reflected in the fact that there was little community dissimilarity between E1 and E5 (Appendix A).

### 4.4. The Co-Occurrence Network Is More Stable during the Plum Rain Season, and the Network Complexity Increases as the Deterministic Process Increases

In natural habitats, microorganisms often do not exist in isolation, but form complex networks with biological interactions [46,47]. Co-occurrence networks had more positive correlations during the plum rain season compared to during the non-plum rain season, reflecting a shift in species interactions from stress-free competition to co-operative symbiosis in response to stress [48]. Previous studies have suggested that the adaptation of one species to environmental pressures may increase the selective pressure on another species, leading to antagonistic coevolution (Red Queen Hypothesis [49]). A higher α-diversity in the non-plum rain season further confirmed that the network was more stable in the non-plum rain season [50]; furthermore, the analysis of the natural connectivity of the network supported the view that the plum rain season led to a decrease in network stability (Figure 3B). Therefore, the plum rain season, a special natural climatic phenomenon, decreased the network stability of the human opportunistic pathogenic bacterial community. Evidence suggests that the keystones of bacterial networks are often subdominant bacteria [51], such as Pseudomonas aeruginosa, Mycobacterium tuberculosis, and Haemophilus influenza, which had a low relative abundance but high biocorrelationships in this study (Appendix A), and the removal of these keystones has the potential to trigger network collapse [52], which could potentially account for the complex but unstable bacterial network of the electric bicycles during the plum rain season.

Furthermore, assembly processes and co-occurrence networks may be intrinsically linked given their co-variation [53], with previous findings confirming that networks tend to be more complex when the contribution of deterministic processes increases [54], and an increase in the complexity of co-occurrence networks as the proportion of deterministic processes in the community increased was also found in our results. However, some previous studies have shown that microbial associations increase with stochastic processes in coastal wetlands or agricultural soils [55,56]. The reason for this discrepancy may be that the present study focused on time scales based on small-scale changes in temperature and humidity, whereas previous studies were mainly based on spatial scales over large distances and latitudes.

### 4.5. Opportunistic Pathogens on Human Daily Necessities Pose Potential Health Risks

In China, respiratory diseases caused by *Haemophilus influenzae* are more prevalent in the summer, while those caused by *Klebsiella pneumoniae* peak in the autumn [57]. Gastrointestinal diseases caused by *Staphylococcus aureus* and *Bacillus cereus* also show a higher incidence in the summer [58,59]. Urinary tract diseases caused by *Enterococcus faecium* and *Proteus mirabilis* peak in the summer as well [60]. The elevated occurrence of these diseases in the summer in the middle and lower reaches of the Yangtze River region in China may be due to the influence of the plum rain season. As in this study, the relative abundance of *Haemophilus influenzae* on washbasins was higher in the plum rain season than that in the non-plum rain season, the relative abundance of *Staphylococcus aureus* on rice was higher in the plum rain season than that in the non-plum rain season, and the relative abundance of *Enterococcus faecium* and *Proteus mirabilis* on cotton cloth was higher in the plum rain season than that in the non-plum rain season (Figure 1C). *Stenotrophomonas maltophilia* is a widespread bacteria in a variety of environments and can even colonize inanimate surfaces [61]. It has been found that *Stenotrophomonas maltophilia* increase with an increasing relative humidity [62], which is consistent with what was observed in this study, that is, the relative abundance of *Stenotrophomonas maltophilia* on daily necessities was higher during the plum rain season than that in the non-plum rain season. *Pseudomonas aeruginosa* has a preference for moist environments and is a common concurrent pathogen with *Stenotrophomonas maltophilia* [63]. Hands [64] and environmental repositories [65] have been shown to be the main sites of *Stenotrophomonas maltophilia* and *Pseudomonas aeruginosa* infections. Therefore, hand contact and use of water may be the transmission route for respiratory diseases caused by *Stenotrophomonas maltophilia*. It has been reported that the numbers of *Staphylococcus aureus*, *Bacillus cereus*, and *Listeria monocytogenes*, which can cause gastrointestinal diseases, are higher in the rainy season than in the dry season [36,66,67]. This reminds us that we should be alert to the health risks posed by *Staphylococcus aureus* in food during the plum rain season and should not ignore the possible health risks posed by other foodborne pathogens. Therefore, it is particularly important to pay attention to food storage and take appropriate measures to prevent bacterial growth during the plum rain season, such as the short-term refrigeration of perishable foods and cooking them before consumption. According to studies, *Enterococcus faecium* and *Proteus mirabilis* are opportunistic human pathogens that cause urinary tract infections and are resistant to a wide range of antibiotics [68,69]; furthermore, the number of *Enterococcus faecium* is significantly higher in the rainy season than in the dry season [70]. It has been shown that the presence of *Enterococcus faecium* on bedding may infect humans and spread to the environment through lint particles [71], posing a potential threat to human health. Therefore, it is necessary to change clothes in time and maintain dryness and cleanliness during the plum rain season to effectively avoid the possibility of bacterial infection. These simple but important measures can reduce the environment for bacterial growth and help to reduce the risk of urinary tract diseases.

## 5. Conclusions

These findings have crucial implications for public health, particularly in the context of climate change. The unique climatic conditions during the plum rain season, characterized by higher temperatures, humidity, and lower air pressure, create an environment conducive to the proliferation of human opportunistic pathogenic bacteria. The increased relative abundance of these pathogens, which are harmful to human health and can cause respiratory, gastrointestinal, and urinary tract diseases through skin contact, ingestion, or inhalation, highlights the potential health risks associated with the plum rain season. Therefore, there is a need to be vigilant and develop management strategies during the plum rain season to reduce potential health risks.

## Figures and Tables

**Figure 1 microorganisms-12-00260-f001:**
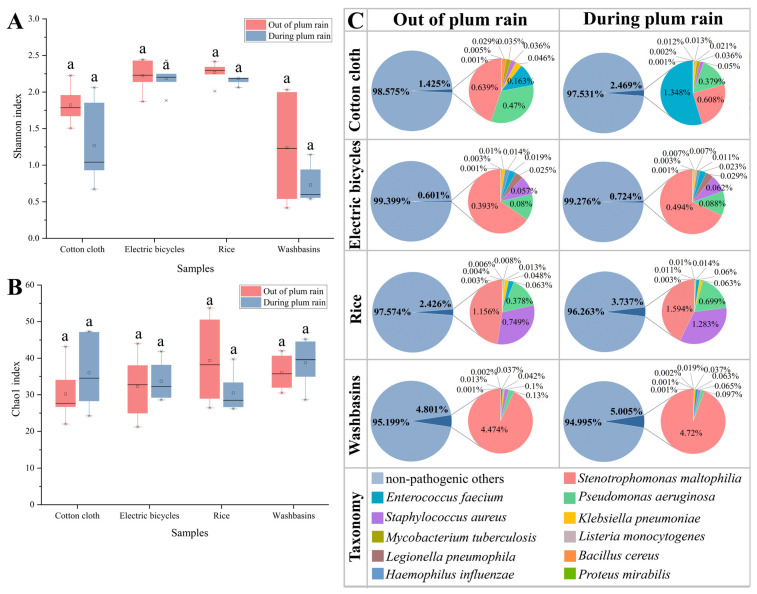
Community diversity and composition of human opportunistic pathogenic bacteria in daily necessities. (**A**) The Shannon index of human opportunistic pathogenic bacterial communities during the non-plum and plum rain seasons; (**B**) the Chao1 index of human opportunistic pathogenic bacterial communities during the non-plum and plum rain seasons, the same lowercase letter indicated that there was no significant difference between the same material in the non-plum and the plum rain season (*p* > 0.05) and (**C**) relative abundance of human opportunistic pathogenic bacteria during the non-plum and plum rain seasons.

**Figure 2 microorganisms-12-00260-f002:**
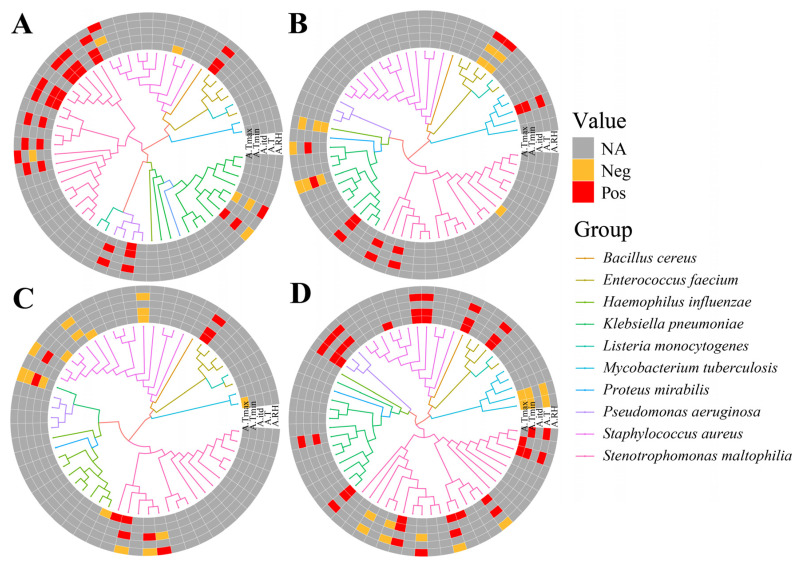
Phylogenetic tree heatmap of human opportunistic pathogenic bacteria associated with environmental factors on daily necessities: (**A**) cotton cloth, (**B**) electric bicycles, (**C**) rice, and (**D**) washbasins. For abbreviations see Table 1.

**Figure 3 microorganisms-12-00260-f003:**
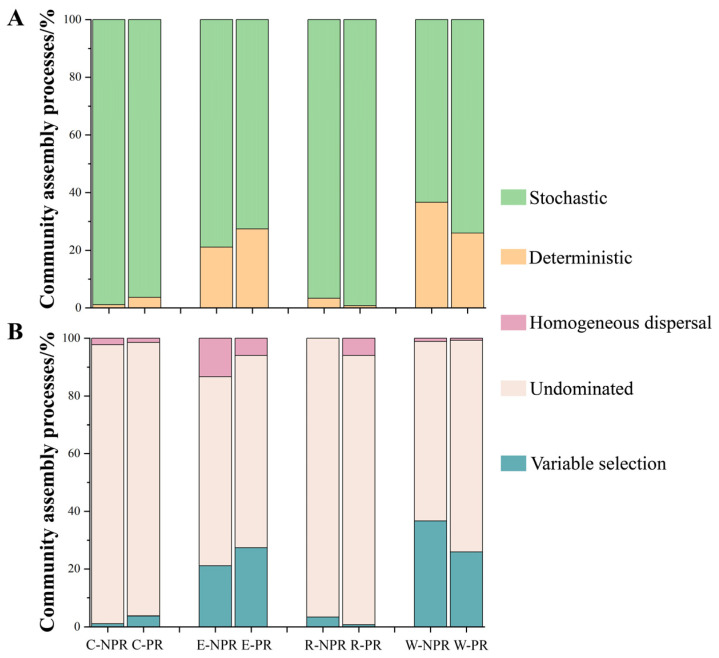
The assembly processes of human opportunistic pathogenic bacterial communities on daily necessities during the non-plum and plum rain seasons. (**A**) Contribution of deterministic and stochastic processes and (**B**) quantitative analysis of the assembly processes.

**Figure 4 microorganisms-12-00260-f004:**
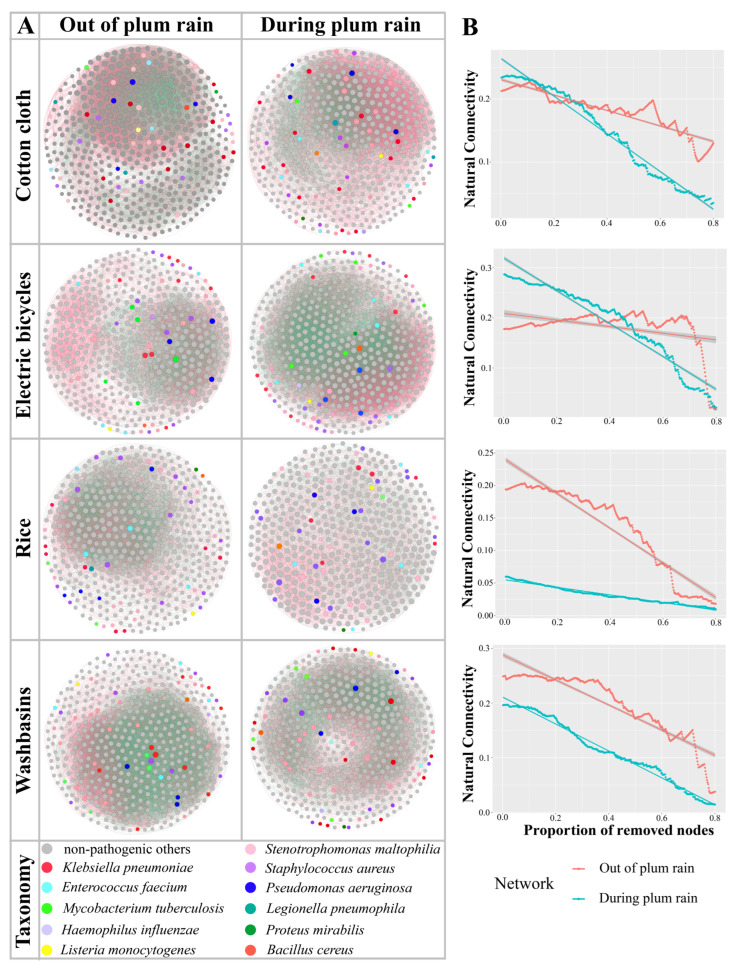
Variations in bacterial co-occurrence networks during the non-plum and plum rain seasons. (**A**) The network was constructed in units of OTUs. The size of the nodes indicates the number of related relationships with other species. Red lines indicate positive correlations and green lines indicate negative correlations. (**B**) Comparison of bacterial network stability of different daily necessities during the non-plum and plum rain seasons, the straight lines are the linear fit to the scatterplots.

**Table 1 microorganisms-12-00260-t001:** Weather indicators at different plum rain stage.

Time Points	Stages	A.Tmax	A.Tmin	A.itd	A.T	A.RH
20 June	Before plum rain	34.28 ± 1.98 b	23.71 ± 1.58 c	10.57 ± 0.90 a	29.00 ± 1.73 c	56.39 ± 5.20 c
27 June	Early of plum rain	33.57 ± 1.50 b	26.86 ± 1.24 ab	6.71 ± 0.45 b	30.21 ± 1.36 bc	71.52 ± 6.06 a
4 July	Middle of plum rain	33.00 ± 1.41 b	26.00 ± 0.53 b	7.00 ± 1.60 b	29.50 ± 0.71 c	70.95 ± 2.57 ab
11 July	Late of plum rain	35.14 ± 1.64 ab	27.86 ± 0.83 a	7.28 ± 1.16 b	31.50 ± 1.16 ab	69.66 ± 4.36 ab
18 July	After plum rain	36.57 ± 2.32 a	28.43 ± 2.26 a	8.14 ± 1.36 b	32.50 ± 2.19 a	63.89 ± 9.29 b

“A.Tmax” represents average maximum temperature, “A.Tmin” represents average minimum temperature, “A.itd” represents average temperature difference, “A.T” represents average temperature, and “A.RH” represents average relative humidity. Different lowercase letters indicate significant differences between different treatments according to one-way ANOVA with Duncan test (*p* < 0.05).

**Table 2 microorganisms-12-00260-t002:** Topological properties of bacterial community networks of different daily necessities between plum rain and non-plum rain seasons.

Topological Indexes	C-NPR	C-PR	E-NPR	E-PR	R-NPR	R-PR	W-NPR	W-PR
Nodes	475	480	472	480	476	469	479	482
Edges	17,069	17,236	13,130	25,298	13,419	6617	20,365	16,661
Positive edges	63.75	67.01	67.01	66.28	57.02	64.33	58.49	59.59
Negative edges	36.25	32.99	32.99	33.72	42.98	35.67	41.51	40.41
Average degree	71.869	71.817	55.636	105.408	56.382	28.157	85.031	69.133
Network diameter	5	6	6	5	5	5	5	5
Graph density	0.152	0.15	0.118	0.22	0.119	0.06	0.178	0.144
Modularity	1.102	0.583	1.36	0.908	1.904	1.309	1.908	2.114
Average path length	2.352	2.229	2.509	2.015	2.517	2.503	2.315	2.198

The “PR” represents “plum rain” and “NPR” represents “non-plum rain”.

## Data Availability

All data supporting the findings of this study are available on request from the corresponding authors (Hui Cao).

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
