# Peer review of "An Assessment of Human Opportunistic Pathogenic Bacteria on Daily Necessities in Nanjing City during Plum Rain Season"

_microorganisms, 2024, doi:10.3390/microorganisms12020260_

Round 1

Reviewer 1 Report

Comments and Suggestions for Authors

The manuscript „Assessment of human opportunistic pathogenic bacteria on daily necessities in Nanjing City during plum rain season”, by Yu X et al approaches an interesting. It starts describing this climatic phenomena, which is a strong asset considering that not everyone is familiar with this. The manuscript has well defined hypothesis.

The materials and methods section is well organized and logically written.

The results section has very interesting results, supported by figures and graphs. It is very interesting how M. tuberculosis is more frequent out of the plum season together with K. pneumoniae while S. aureus and P. aeruginosa during the rainy season.

Discussion:

Have you considered adding a few paragraphs on the increasing number of diseases with some pathogens in each season? For example in my country we have more cases of respiratory P. aeruginosa infection during winter season in COPD patients. Maybe you could add a phrase if you have data on this matter in your country.

I would suggest to create a Conclusion section at the end of the manuscript, where you could move the last paragraph of the manuscript from Discussion. It is fine as you have written it as well, but this way it would highlight the results and probably also add a sentence in the Conclusion on the most frequent pathogens in different seasons.

Author Response

Comment 1 Have you considered adding a few paragraphs on the increasing number of diseases with some pathogens in each season? For example in my country we have more cases of respiratory P. aeruginosa infection during winter season in COPD patients. Maybe you could add a phrase if you have data on this matter in your country.

Answer: Thank you for your comment, the incidence of respiratory, gastrointestinal and urinary tract diseases caused by bacteria in this country has been added to the Section Discussion.

Comment 2 I would suggest to create a Conclusion section at the end of the manuscript, where you could move the last paragraph of the manuscript from Discussion. It is fine as you have written it as well, but this way it would highlight the results and probably also add a sentence in the Conclusion on the most frequent pathogens in different seasons.

Answer: Thank you very much for your reminder, the Section "Conclusion" has been added. Since this study only examined the characteristics of human opportunistic pathogenic bacterial communities on daily necessities in the non-plum rain season and plum rain season, the most common pathogens in the other seasons were not examined.

We hereby resubmit the revised manuscript and hope that all corrections are satisfactory. We look forward to your decision.

Reviewer 2 Report

Comments and Suggestions for Authors

The manuscript titled “Assessment of human opportunistic pathogenic bacteria on daily necessities in Nanjing City during plum rain season” is devoted to studу of human opportunistic pathogenic bacteria communities  during the plum rain season. In this study, the characteristics of human opportunistic pathogenic bacterial communities in daily necessities during the non-plum and plum rain seasons were investigated using high-throughput sequencing technology. The results revealed that the relative abundance of human opportunistic pathogenic bacteria was higher in the plum rain season than in the non-plum rain season. Both temperature and relative humidity affected human opportunistic pathogenic bacterial communities. Stochastic processes dominated the assembly process of human opportunistic pathogenic bacterial communities, and undominated processes prevailed. This study aims to raise awareness of human opportunistic pathogenic bacterial communities in daily necessity and calls for better public health management during the plum rain season.

This work is reporting high quality sequencing data and well-written, but it is based on one-year experiment and the conclusions should be treated as preliminary.  

Besides this, there are some other  notes:

Line 30-64 Note: The introduction part from line 30t to 64 and reference s 1- 17 are devoted to general information about plum rain season without relation to opportunistic pathogens found in air. It is better to describe other studies devoted to air analysis.

Line 98 2.1 Study site

Note: Add average temperature and humidity for the plum rain season comparing to other seasons.

Line 108 Daily necessities including cotton cloth (C), electric bicycles (E), rice (R) and washbasins in dormitory (W) were sampled

Note: Please, describe the reasons for choice of  C, E, R and W for analysis.

Line 237 Figure 1. Community diversity and composition of human opportunistic pathogenic bacteria in daily necessities. (A) The Shannon index of human opportunistic pathogenic bacterial communities 238 during the non-plum and plum rain seasons;

Note: Results are overlapping for cotton cloth, electric bicycles and washing basins.

Line 238. (B) the Chao1 index of human opportunistic pathogenic bacterial communities during the non-plum and plum rain seasons;

Note: Results are overlapping for cotton cloth, electric bicycles, rice,  and washing basins.

Line 239 (C) relative abundance of human opportunistic pathogenic bacteria during the non-plum and plum rain seasons.

Note: Results are different for cotton cloth, but not for electric bicycles, rice,  and washing basins.

Line 244 Human opportunistic pathogenic bacteria in washbasins showed the highest susceptibility to the combined effects of temperature and relative humidity, followed by human opportunistic pathogenic bacteria in cotton cloth, electric bicycles, and rice (Figure 2A-D).

Note:  This conclusion is made based on 5 collection points (Table 1 20-Jun, 27-Jun, 4-Jul, 11-Jul, and 18-Jul). It cannot be accepted as reliable and repeatable result.

Comments on the Quality of English Language

Minor editing of English language required

Author Response

Comment 1 Line 30-64

Note: The introduction part from line 30t to 64 and reference s 1- 17 are devoted to general information about plum rain season without relation to opportunistic pathogens found in air. It is better to describe other studies devoted to air analysis.

Answer: Thank you for your suggestion, the Section Introduction of the manuscript has been revised to remove the content on the causes of the plum rain season and add the relationship between public health diseases and seasonality.

Comment 2 Line 98 2.1 Study site

Note: Add average temperature and humidity for the plum rain season comparing to other seasons.

Answer: Thank you for your kind recommendation, the average temperature, average relative humidity and precipitation for the plum rain season and all seasons of the year have been tabulated and added to the Supplementary materials, as can be seen in Table S2, the average temperature, average relative humidity and precipitation were higher in the plum rain season than in the other seasons, except that the average temperature was slightly lower than the summer average temperature.

Comment 3 Line 108 Daily necessities including cotton cloth (C), electric bicycles (E), rice (R) and washbasins in dormitory (W) were sampled

Note: Please, describe the reasons for choice of  C, E, R and W for analysis.

Answer: Thank you for pointing out, this study is focused on the human opportunistic pathogenic bacterial communities on daily necessities in Nanjing City during the plum rain season. Four daily necessities, namely cotton cloth, rice, washbasins, and electric bicycles, were selected from the four aspects of clothing, food, housing, and transportation, which are closely related to human activities.

Comment 4 Line 237 Figure 1. Community diversity and composition of human opportunistic pathogenic bacteria in daily necessities. (A) The Shannon index of human opportunistic pathogenic bacterial communities 238 during the non-plum and plum rain seasons;

Note: Results are overlapping for cotton cloth, electric bicycles and washing basins.

Answer: Thank you for your question. The box plot in Figure 1A represents the Shannon index of human opportunistic pathogenic bacteria on daily necessities during the non-plum rain season and the plum rain season, and the horizontal lines from top to bottom represent the upper margin, upper quartile, median, lower quartile, and lower margin in that order, and we determine the magnitude of the Shannon index by comparing the magnitude of the median. Previously, when some scholars studied the change rule of soil bacterial community in a cadmium-polluted farmland ecosystem under severe and long-term cadmium pollution, the median was used to represent the size of the index in the box line diagrams for Shannon index and Chao1 index of bacterial community [1].

Comment 5 Line 238. (B) the Chao1 index of human opportunistic pathogenic bacterial communities during the non-plum and plum rain seasons;

Note: Results are overlapping for cotton cloth, electric bicycles, rice,  and washing basins.

Answer: Thank you for your question. The box plot in Figure 1B represents the Chao1 index of human opportunistic pathogenic bacteria on daily necessities during the non-plum rain season and the plum rain season, and the horizontal lines from top to bottom represent the upper margin, upper quartile, median, lower quartile, and lower margin in that order, and we determine the magnitude of the Chao1 index by comparing the magnitude of the median. When some scholars studied the microbial diversity and community composition characteristics of groundwater in the salt-freshwater transition zone, the median comparison size was also taken in the α-diversity box plots for bacterial communities [2].

Comment 6 Line 239 (C) relative abundance of human opportunistic pathogenic bacteria during the non-plum and plum rain seasons.

Note: Results are different for cotton cloth, but not for electric bicycles, rice,  and washing basins.

Answer: Thank you for your question, the Section Results 3.2 has been modified to express relative abundance in more detail and with more specificity, from Figure 1C, it can be found that the relative abundance of human opportunistic pathogenic bacteria on cotton cloth, electric bicycles, rice and washbasins during the non-plum rain season was 1.425%, 0.601%, 2.426% and 4.801% respectively, and the relative abundance of human opportunistic pathogenic bacteria on cotton cloth, electric bicycles, rice and washbasins during the plum rain season was 2.469%, 0.724%, 3.737% and 5.005%. On all daily necessities, the relative abundance of human opportunistic pathogenic bacteria was higher during the plum rain season than during the non-plum rain season.

Comment 7 Line 244 Human opportunistic pathogenic bacteria in washbasins showed the highest susceptibility to the combined effects of temperature and relative humidity, followed by human opportunistic pathogenic bacteria in cotton cloth, electric bicycles, and rice (Figure 2A-D).

Note:  This conclusion is made based on 5 collection points (Table 1 20-Jun, 27-Jun, 4-Jul, 11-Jul, and 18-Jul). It cannot be accepted as reliable and repeatable result.

Answer: Thank you for pointing out, the data from the five collection points was tabulated by collating climate data from the seven days prior to sampling, which is the result of the impact of climatic conditions on daily necessities during the 2022 Nanjing plum rain season. We apologize for our lack of clarity in the Section Materials and Methods and have revised and added to the Section Materials and Methods of the original manuscript so that the results as reliable and accurate.

Thank you for your valuable and thoughtful comments. We have carefully checked and improved the English writing in the revised manuscript. We hereby resubmit the revised manuscript and hope that all corrections are satisfactory. We look forward to your decision.

  1. Deng, Y.; Fu, S.; Sarkodie, E.K.; Zhang, S.; Jiang, L.; Liang, Y.; Yin, H.; Bai, L.; Liu, X.; Liu, H.; et al. Ecological responses of bacterial assembly and functions to steep Cd gradient in a typical Cd-contaminated farmland ecosystem. Ecotoxicology and Environmental Safety 2022, 229, 113067, doi:https://doi.org/10.1016/j.ecoenv.2021.113067.
  2. Chen, L.; Hu, B.X.; Dai, H.; Zhang, X.; Xia, C.-A.; Zhang, J. Characterizing microbial diversity and community composition of groundwater in a salt-freshwater transition zone. Science of The Total Environment 2019, 678, 574-584, doi:https://doi.org/10.1016/j.scitotenv.2019.05.017.

Reviewer 3 Report

Comments and Suggestions for Authors

The manuscript is well written. The abstract should include the main findings of the research using actual numbers. Please do not state the aim of the study in the Abstract section. How many replicate DNA samples were extracted for each C,E,R,W? In Section 3.2 please include the relative abundances. Also, be more specific on the observed shift regarding the relative abundances during variable conditions (i.e. lines 247-250). Please include a distinct section with the conclusions of the research.

Author Response

Comment 1 The abstract should include the main findings of the research using actual numbers. Please do not state the aim of the study in the Abstract section.

Answer: Thank you for your comment, the Section Abstract has been revised to include the relative abundance of human opportunistic pathogenic bacteria on daily necessities during the non-plum rain season and the plum rain season.

Comment 2 How many replicate DNA samples were extracted for each C,E,R,W?

Answer: Thank you for your question, as described in the Section 2.2 Experimental Design and Sampling of manuscript, the four daily necessities, cotton cloth, electric bicycles, rice and washbasins, were sampled at five points in time, with a total of three replicates per sampling, making a total of 60 samples.

Comment 3 In Section 3.2 please include the relative abundances. Be more specific on the observed shift regarding the relative abundances during variable conditions (i.e. lines 247-250).

Answer: Thank you for your comment, revisions and additions have been made to the Section 3.2 of the manuscript.

Comment 4 Please include a distinct section with the conclusions of the research.

Answer: Thank you for your suggestion, the Section “Conclusion” has been added.

Thank you very much for taking the time to review this manuscript. We hereby resubmit the revised manuscript and hope that all corrections are satisfactory. We look forward to your decision.

Reviewer 4 Report

Comments and Suggestions for Authors

This research paper presents opportunistic pathogenic bacteria collected from four different types of daily necessities during the non-plum and plum rain seasons by means of genetic analysis. This is an interesting research topic because there is little information on pathogenic bacteria attached to daily necessities, which may support the fact that infectious diseases are more frequent during the rainy season. However, the paper lacks an explanation of the important points of the setting conditions. In addition, there is a lack of data and scientific evidence on the relationship and consistency between weather conditions and exposure environments, and some of the considerations are arbitrary.

The following are the major points raised:

(1) Reasons for the selection of four types of daily necessities (cotton cloth, electric bicycles, rice, and washbasin) are necessary.

(2) Information on the exposure environment of each daily necessities is needed.

(3) Data and scientific evidence on the relationship and consistency between temperature and humidity in the meteorological data and the exposure environment of each sample are needed.

(4) The appropriateness of statistically analyzing the data of three rainy season data groups (three surveys), one group before the plum rain seasons (one survey), and one group after the plum rain seasons (one survey) for the non-plum and plum rain seasons weather conditions should be explained.

(5) All the obtained genetic information should be registered in a gene bank that has been authorized.

The specific points are as follows:

(1) Lines 108-109

The reason why four samples with different materials, purposes of use, and places of use were selected should be explained in detail. Information on each individual sample should also be explained. In addition, information on the sampling location and exposure environment to the atmosphere should be added.

(2) Lines 113-114

It should be indicated that the location of each sample is guaranteed to be related to the atmospheric temperature and humidity, which are meteorological conditions. The air dust and rain samples in the previous reports and the samples in this study are totally different.

(3) Section 3.2

The number of OUT reads obtained and the bacterial flora analysis database should be shown as supplemental data.

(4) Lines 259-260

If indoor and outdoor effects occur, the comparison should be based on indoor and outdoor temperature and humidity data, not weather data. This is the most important issue for this paper in general.

(5) Lines 326-328

Bacterial attachment, growth, and survival are significantly affected by the exposure environment of cloth. Since information on indoor and outdoor atmospheric environment is missing, we have to conclude that this is an arbitrary consideration.

(6) Line 334-335

The number and flora of pathogens are significantly affected by the method and frequency of using the washbasin, the water used, and the wetting and drying of the surface. Consideration based on these scientific evidences is necessary.

(7) Section 4.2

Again, the validity of the temperature and humidity data of the meteorological information and the conditions of the exposure environment of each sample should be shown to be consistent. Information on the atmospheric environment to which the samples of completely different subsistence products are exposed is essential.

Author Response

The following are the major points raised:

Comment 1 Reasons for the selection of four types of daily necessities (cotton cloth, electric bicycles, rice, and washbasin) are necessary.

Answer: Thank you for pointing out, this study is focused on the human opportunistic pathogenic bacterial communities on daily necessities in Nanjing City during the plum rain season. Four daily necessities, namely cotton cloth, rice, washbasins, and electric bicycles, were selected from the four aspects of clothing, food, housing, and transportation, which are closely related to human activities, respectively.

Comment 2 Information on the exposure environment of each daily necessities is needed.

Answer: Thank you very much for pointing this out, changes and additions have been made to the manuscript Section 2.2 Experimental design and sampling. In this study, the cotton cloth, electric bicycles, rice and washbasins were in an open environment, the washbasins were located in the dormitory balconies directly exposed to the atmosphere, the balconies were open to the outdoors without obstacles, the electric bicycles were located in a completely outdoor environment. We are very sorry for the lack of clarity of expression in the original manuscript, we have revised the materials and methods in the original manuscript, thank you again for pointing out.

Comment 3 Data and scientific evidence on the relationship and consistency between temperature and humidity in the meteorological data and the exposure environment of each sample are needed.

Answer: Before we conducted our study, we had reviewed the relevant literature, one of which, published in Environment International, studied the characteristics of airborne bacterial communities in indoor and outdoor environments during consecutive haze events in Beijing, and in their study, the balconies were open to the outdoor environment and were directly exposed to the atmospheric environment, so they regarded them as outdoor environments [1], in addition, an article published in Microbiome recorded air temperature and relative humidity to explore seasonal changes in subway air and surface bacterial communities [2], all of which provided insights and scientific evidence for our study.

Comment 4 The appropriateness of statistically analyzing the data of three rainy season data groups (three surveys), one group before the plum rain seasons (one survey), and one group after the plum rain seasons (one survey) for the non-plum and plum rain seasons weather conditions should be explained.

Answer: The plum rain season in Nanjing in 2022 Lasted from 23 June to 11 July, the climate information of the sampling time point of this study is the integration of the information of the previous seven days of the sampling point, and before the entry into the plum and after the plum out of the plum are both belong to the non-plum rain period, so the whole time period is divided into the non-plum rain season and the plum rain season. In 2017, the national standard GB/T 33671-2017 for plum rain detection indicators was formulated, and the plum rain monitoring indicators have detailed expressions and objective provisions for the time of entering (leaving) the plum and the intensity of the plum rain, etc. There are three important indicators: the location of the subtropical high pressure ridgeline, the condition of the average daily temperature and the judgement of the number of rainy days during the rainy period. During the plum rain season, the location of the 5-day slide of the subtropical ridge needs to meet the following conditions (see table below). When the 5-day sliding position of the sub-high ridgeline exceeds the northern boundary position by 2 degrees of latitude and no rainy days continue to occur, and hot and dry weather occurs in the monitoring region, the plum rain season in the region ends. Regional plum rain needs to be considered in the entry conditions for plum rain to occur in a hot and humid environment with a mean daily temperature ≥ 22℃.

Table Range of activity of the sub-high ridgeline during the plum rain season

Regional Selection

Southern Border

North Border

Jiangnan District

≥ 18°N

< 25°N

Middle and Lower Yangtze River Region

≥ 19°N

< 26°N

Jianghuai District

≥ 20°N

< 27°N

Comment 5 All the obtained genetic information should be registered in a gene bank that has been authorized.

Answer: Thank you for the reminder, all data have been uploaded to the NCBI database with the following repository/repository names and login numbers: Genomic sequencing data have been deposited in the NCBI Sequence Read Archive (BioProject ID PRJNA1019093, Submission SUB13852553).

The specific points are as follows:

Comment 1 Lines 108-109

The reason why four samples with different materials, purposes of use, and places of use were selected should be explained in detail. Information on each individual sample should also be explained. In addition, information on the sampling location and exposure environment to the atmosphere should be added.

Answer: Thank you for your comment. This study is based on the human opportunistic pathogenic bacterial communities on daily necessities in Nanjing during the plum rain season. Four daily necessities, namely cotton cloth, electric bicycles, rice and washbasins, were selected from the four aspects of clothing, transportation, food, and housing, which are closely related to human activities, respectively. Cotton cloth and rice were located in an open environment, and the washbasins in dormitories were located in the balconies, which were open to the outdoors without obstacles, and the electric bicycles were in the completely outdoor environment, sorry for the lack of clarity in the original manuscript, which have been added to the Section 2.2 Experimental Design and Sampling.

Comment 2 It should be indicated that the location of each sample is guaranteed to be related to the atmospheric temperature and humidity, which are meteorological conditions. The air dust and rain samples in the previous reports and the samples in this study are totally different.

Answer: Thank you for pointing this out. A previously published article in Microbiome examined the dynamics of the underground microbiome and a direct comparison of air and surface bacterial communities, where they recorded air temperature and humidity, applied jointly to both air and surface bacterial communities [2], just as all daily necessities in this study were exposed to atmospheric conditions and were jointly and directly affected by temperature and relative humidity.

Comment 3 Section 3.2

The number of OUT reads obtained and the bacterial flora analysis database should be shown as supplemental data.

Answer: Thank you for your suggestion, the OTU reads have been added to the Section Results, the databases used for the analysis of this study were Silva and Enhanced Infectious Disease databases, as mentioned in the Section Material and methods.

Comment 4 Lines 259-260

If indoor and outdoor effects occur, the comparison should be based on indoor and outdoor temperature and humidity data, not weather data. This is the most important issue for this paper in general.

Answer: Thank you very much for pointing this out, changes and additions have been made to the manuscript Section 2.2 Experimental design and sampling. In this study, all daily necessities were in an open environment, the washbasins were located in the dormitory balconies directly exposed to the atmosphere, the balconies were open to the outdoors without obstacles, the electric bicycles were located in a completely outdoor environment, I'm very sorry for the lack of clarity of expression in the original manuscript, I have revised the materials and methods in the original manuscript, thank you again for pointing out.

Comment 5 Lines 326-328

Bacterial attachment, growth, and survival are significantly affected by the exposure environment of cloth. Since information on indoor and outdoor atmospheric environment is missing, we have to conclude that this is an arbitrary consideration.

Answer: Thank you for your question, cotton cloth was in an open environment and was directly affected by atmospheric factors.

Comment 6 Line 334-335

The number and flora of pathogens are significantly affected by the method and frequency of using the washbasin, the water used, and the wetting and drying of the surface. Consideration based on these scientific evidences is necessary.

Answer: Thank you for mentioning, we also considered this problem, so that the washbasins were considered to be subject to human interference, in the sampling process, we tried to keep the washbasins basically unchanged, the frequency of use to maintain roughly the same situation, and that each time a sample was taken, a sterile sampling plate was placed on the surface of the dry washbasin, and the sample was taken with a sterile swab that had been moistened with sterile water.

Comment 7 Section 4.2

Again, the validity of the temperature and humidity data of the meteorological information and the conditions of the exposure environment of each sample should be shown to be consistent. Information on the atmospheric environment to which the samples of completely different subsistence products are exposed is essential.

Answer: Thank you for your proposal, all daily necessities in this study were in an open environment and were directly affected by the atmosphere, this study mainly focused on the effect of the same meteorological factors of the particular climatic phenomenon of the plum rain season on the human opportunistic pathogenic bacterial communities on different daily necessities, so the meteorological factors were used as the environmental factors.

Thank you very much for taking the time to review this manuscript. We hereby resubmit the revised manuscript and hope that all corrections are satisfactory. We look forward to your decision.

  1. Guo, J.; Xiong, Y.; Shi, C.; Liu, C.; Li, H.; Qian, H.; Sun, Z.; Qin, C. Characteristics of airborne bacterial communities in indoor and outdoor environments during continuous haze events in Beijing: Implications for health care. Environment International 2020, 139, 105721, doi:https://doi.org/10.1016/j.envint.2020.105721.
  2. Gohli, J.; Bøifot, K.O.; Moen, L.V.; Pastuszek, P.; Skogan, G.; Udekwu, K.I.; Dybwad, M. The subway microbiome: seasonal dynamics and direct comparison of air and surface bacterial communities. Microbiome 2019, 7, 160, doi:10.1186/s40168-019-0772-9.

Round 2

Reviewer 2 Report

Comments and Suggestions for Authors

The manuscript titled “Assessment of human opportunistic pathogenic bacteria on daily necessities in Nanjing City during plum rain season” is devoted to studу of human opportunistic pathogenic bacteria communities  during the plum rain season. In this study, the characteristics of human opportunistic pathogenic bacterial communities in daily necessities during the non-plum and plum rain seasons were investigated using high-throughput sequencing technology. The results revealed that the relative abundance of human opportunistic pathogenic bacteria was higher in the plum rain season than in the non-plum rain season. Both temperature and relative humidity affected human opportunistic pathogenic bacterial communities. Stochastic processes dominated the assembly process of human opportunistic pathogenic bacterial communities, and undominated processes prevailed. This study aims to raise awareness of human opportunistic pathogenic bacterial communities in daily necessity and calls for better public health management during the plum rain season.

This work is reporting high quality sequencing data and well-written, Authors improved the manuscript according to Reviewer's notes. 

Reviewer 4 Report

Comments and Suggestions for Authors

My key doubts were resolved because I could confirm that all four daily necessities were exposed in the atmospheric environment and still monitored on a continuous basis.

The revisions, content, and response are appropriate.

Therefore, I judge that the revised paper is worthy of acceptance for publication.